# Data driven estimation of Laplace-Beltrami operator

**Frédéric Chazal**
Inria Saclay
Palaiseau France
frederic.chazal@inria.fr

**Ilaria Giulini**
Inria Saclay
Palaiseau France
ilaria.giulini@me.com

**Bertrand Michel**
Ecole Centrale de Nantes
Laboratoire de Mathématiques Jean Leray (UMR 6629 CNRS)
Nantes France
bertrand.michel@ec-nantes.fr

## Abstract

Approximations of Laplace-Beltrami operators on manifolds through graph Laplacians have become popular tools in data analysis and machine learning. These discretized operators usually depend on bandwidth parameters whose tuning remains a theoretical and practical problem. In this paper, we address this problem for the unnormalized graph Laplacian by establishing an oracle inequality that opens the door to a well-founded data-driven procedure for the bandwidth selection. Our approach relies on recent results by Lacour and Massart [LM15] on the so-called Lepski's method.

## 1 Introduction

The Laplace-Beltrami operator is a fundamental and widely studied mathematical tool carrying a lot of intrinsic topological and geometric information about the Riemannian manifold on which it is defined. Its various discretizations, through graph Laplacians, have inspired many applications in data analysis and machine learning and led to popular tools such as Laplacian EigenMaps [BN03] for dimensionality reduction, spectral clustering [VL07], or semi-supervised learning [BN04], just to name a few.

During the last fifteen years, many efforts, leading to a vast literature, have been made to understand the convergence of graph Laplacian operators built on top of (random) finite samples to Laplace-Beltrami operators. For example pointwise convergence results have been obtained in [BN05] (see also [BN08]) and [HAL07], and a (uniform) functional central limit theorem has been established in [GK06]. Spectral convergence results have also been proved by [BN07] and [VLBB08]. More recently, [THJ11] analyzed the asymptotic of a large family of graph Laplacian operators by taking the diffusion process approach previously proposed in [NLCK06].

Graph Laplacians depend on scale or bandwidth parameters whose choice is often left to the user. Although many convergence results for various metrics have been established, little is known about how to rigorously and efficiently tune these parameters in practice. In this paper we address this problem in the case of unnormalized graph Laplacian. More precisely, given a Riemannian manifold $M$ of known dimension $d$ and a function $f : M \to \mathbb{R}$, we consider the standard unnormalized graph Laplacian operator defined by

$$\hat{\Delta}_h f(y) = \frac{1}{nh^{d+2}} \sum_i K\left(\frac{y - X_i}{h}\right)[f(X_i) - f(y)], \qquad y \in M,$$

where $h$ is a bandwidth, $X_1, \ldots, X_n$ is a finite point cloud sampled on $M$ on which the values of $f$ can be computed, and $K$ is the Gaussian kernel: for $y \in \mathbb{R}^m$

$$K(y) = \frac{1}{(4\pi)^{d/2}} e^{-\|y\|_m^2/4}, \tag{1}$$

where $\|y\|_m$ is the Euclidean norm in the ambient space $\mathbb{R}^m$.

In this case, previous results (see for instance [GK06]) typically say that the bandwidth parameter $h$ in $\hat{\Delta}_h$ should be taken of the order of $n^{-\frac{1}{d+2+\alpha}}$ for some $\alpha > 0$, but in practice, for a given point cloud, these asymptotic results are not sufficient to choose $h$ efficiently. In the context of neighbor graphs [THJ11] proposes self-tuning graphs by choosing $h$ locally in terms of the distances to the $k$-nearest neighbor, but note that $k$ still need to be chosen and moreover as far as we know there is no guarantee for such method to be rate-optimal. More recently a data driven method for spectral clustering has been proposed in [Rie15]. Cross validation [AC$^+$10] is the standard approach for tuning parameters in statistics and machine learning. Nevertheless, the problem of choosing $h$ in $\hat{\Delta}_h$ is not easy to rewrite as a cross validation problem, in particular because there is no obvious contrast corresponding to the problem (see [AC$^+$10]).

The so-called Lepski's method is another popular method for selecting the smoothing parameter of an estimator. The method has been introduced by Lepski [Lep92b, Lep93, Lep92a] for kernel estimators and local polynomials for various risks and several improvements of the method have then been proposed, see [LMS97, GL09, GL$^+$08]. In this paper we adapt Lepski's method for selecting $h$ in the graph Laplacian estimator $\hat{\Delta}_h$. Our method is supported by mathematical guarantees: first we obtain an oracle inequality - see Theorem 3.1 - and second we obtain the correct rate of convergence - see Theorem 3.3 - already proved in the asymptotical studies of [BN05] and [GK06] for non data-driven choices of the bandwidth. Our approach follows the ideas recently proposed in [LM15], but for the specific problem of Laplacian operators on smooth manifolds. In this first work about the data-driven estimation of Laplace-Beltrami operator, we focus as in [BN05] and [GK06] on the pointwise estimation problem: we consider a smooth function $f$ on $M$ and the aim is to estimate $\hat{\Delta}_h f$ for the $L^2$-norm $\|\cdot\|_{2,M}$ on $M \subset \mathbb{R}^m$. The data driven method presented here may be adapted and generalized for other types of risks (uniform norms on functional family and convergence of the spectrum) and other types of graph Laplacian operators, this will be the subject of future works.

The paper is organized as follows: Lepski's method is introduced in Section 2. The main results are stated in Section 3 and a sketch of their proof is given in Section 4 (the complete proofs are given in the supplementary material). A numerical illustration and a discussion about the proposed method are given in Sections 5 and 6 respectively.

## 2 Lepski's procedure for estimating the Laplace-Beltrami operator

All the Riemannian manifolds considered in the paper are smooth compact $d$-dimensional submanifolds (without boundary) of $\mathbb{R}^m$ endowed with the Riemannian metric induced by the Euclidean structure of $\mathbb{R}^m$. Recall that, given a compact $d$-dimensional smooth Riemannian manifold $M$ with volume measure $\mu$, its Laplace-Beltrami operator is the linear operator $\Delta$ defined on the space of smooth functions on $M$ as $\Delta(f) = -\operatorname{div}(\nabla f)$ where $\nabla f$ is the gradient vector field and $\operatorname{div}$ the divergence operator. In other words, using the Stoke's formula, $\Delta$ is the unique linear operator satisfying

$$\int_M \|\nabla f\|^2 d\mu = \int_M \Delta(f) f d\mu.$$

Replacing the volume measure $\mu$ by a distribution P which is absolutely continuous with respect to $\mu$, the weighted Laplace-Beltrami operator $\Delta_P$ is defined as

$$\Delta_P f = \Delta f + \frac{1}{p} \langle \nabla p, \nabla f \rangle, \tag{2}$$

where $p$ is the density of P with respect to $\mu$. The reader may refer to classical textbooks such as, e.g., [Ros97] or [Gri09] for a general and detailed introduction to Laplace operators on manifolds.

In the following, we assume that we are given n points $X_1, \ldots, X_n$ sampled on $M$ according to the distribution P. Given a smooth function $f$ on $M$, the aim is to estimate $\Delta_P f$, by selecting

an estimator in a given finite family of graph Laplacian $(\hat{\Delta}_h f)_{h \in \mathcal{H}}$, where $\mathcal{H}$ is a finite family of bandwidth parameters.

Lepski's procedure is generally presented as a method for selecting bandwidth in an adaptive way. More generally, this method can be seen as an estimator selection procedure.

## 2.1 Lepski's procedure

We first shortly explain the ideas of Lepski's method. Consider a target quantity $s$, a collection of estimators $(\hat{s}_h)_{h \in \mathcal{H}}$ and a loss function $\ell(\cdot, \cdot)$. A standard objective when selecting $\hat{s}_h$ is trying to minimize the risk $\mathbb{E}\ell(s, \hat{s}_h)$ among the family of estimators. In most settings, the risk of an estimator can be decomposed into a bias part and a variance part. Of course neither the risk, the bias nor the variance of an estimator are known in practice. However in many cases, the variance term can be controlled quite precisely. Lepski's method requires that the variance of each estimator $\hat{s}_h$ can be tightly upper bounded by a quantity $v(h)$. In most cases, the bias can be written as $\ell(s, \bar{s}_h)$ where $\bar{s}_h$ corresponds to some (deterministic) averaged version of $\hat{s}_h$. It thus seems natural to estimate $\ell(s, \bar{s}_h)$ by $\ell(\hat{s}_{h'}, \hat{s}_h)$ for some $h'$ smaller than $h$. The later quantity incorporates some randomness while the bias does not. The idea is to remove the "random part" of the estimation by considering $[\ell(\hat{s}_{h'}, \hat{s}_h) - v(h) - v(h')]_+$, where $[\,]_+$ denotes the positive part. The bias term is estimated by considering all pairs of estimators $(s_h, \hat{s}_{h'})$ through the quantity $\sup_{h' \leq h} [\ell(\hat{s}_{h'}, \hat{s}_h) - v(h) - v(h')]_+$. Finally, the estimator minimizing the sum of the estimated bias and variance is selected, see eq. (3) below.

In our setting, the control of the variance of the graph Laplacian estimators $\hat{\Delta}_h$ is not tight enough to directly apply the above described method. To overcome this issue, we use a more flexible version of Lepski's method that involves some multiplicative coefficients $a$ and $b$ introduced in the variance and bias terms. More precisely, let $V(h) = V_f(h)$ be an upper bound for $\mathbb{E}[\|(\mathbb{E}[\hat{\Delta}_h] - \hat{\Delta}_h)f\|_{2,M}^2]$. The bandwidth $\hat{h}$ selected by our Lepski's procedure is defined by

$$\hat{h} = \hat{h}_f = \arg\min_{h \in \mathcal{H}} \{B(h) + bV(h)\} \tag{3}$$

where

$$B(h) = B_f(h) = \max_{h' \leq h, \, h' \in \mathcal{H}} \left[ \|(\hat{\Delta}_{h'} - \hat{\Delta}_h)f\|_{2,M}^2 - aV(h') \right]_+ \tag{4}$$

with $0 < a \leq b$. The calibration of the constants $a$ and $b$ in practice is beyond the scope of this paper, but we suggest a heuristic procedure inspired from [LM15] in section 5.

## 2.2 Variance of the graph Laplacian for smooth functions

In order to control the variance term, we consider for this paper the set $\mathcal{F}$ of smooth functions $f : M \to \mathbb{R}$ uniformly bounded up to the third order. For some constant $C_\mathcal{F} > 0$, let

$$\mathcal{F} = \left\{ f \in \mathcal{C}^3(M, \mathbb{R}), \, \|f^{(k)}\|_\infty \leq C_\mathcal{F}, \, k = 0, \ldots, 3 \right\}. \tag{5}$$

We introduce some notation before giving the variance term for $f \in \mathcal{F}$. Define

$$D_\alpha = \frac{1}{(4\pi)^d} \int_{\mathbb{R}^d} \left( \frac{C\|u\|_d^{\alpha+2}}{2} + C_1\|u\|_d^\alpha \right) e^{-\|u\|_d^2/4} \, \mathrm{d}u \tag{6}$$

$$\tilde{D}_\alpha = \frac{1}{(4\pi)^{d/2}} \int_{\mathbb{R}^d} \left( \frac{C\|u\|_d^{\alpha+2}}{4} + C_1\|u\|_d^\alpha \right) e^{-\|u\|_d^2/8} \, \mathrm{d}u \tag{7}$$

where $C$ and $C_1$ are geometric constants that only depend on the metric structure of $M$ (see Appendix). We also introduce the $d$-dimensional Gaussian kernel on $\mathbb{R}^d$:

$$K_d(u) = \frac{1}{(4\pi)^{d/2}} e^{-\|u\|_d^2/4}, \qquad u \in \mathbb{R}^d$$

and we denote by $\|\cdot\|_{p,d}$ the $L^p$-norm on $\mathbb{R}^d$. The next proposition provides an explicit bound $V(h)$ on the variance term.

**Proposition 2.1.** *Given $h \in \mathcal{H}$, for any $f \in \mathcal{F}$, we have*

$$V(h) = \frac{C_{\mathcal{F}}^2}{nh^{d+2}}\left(2\omega_d \|K_d\|_{2,d}^2 + \alpha_d(h)\right),$$

*where*

$$\alpha_d(h) = h^2\left(2D_4 + D + 3\omega_d \|K_d\|_{2,d}^2\right) + h^4 \frac{D_6 + 3D}{2} \tag{8}$$

*with*

$$D = \frac{3\mu(M)}{(4\pi)^{d/2}} \quad and \quad \omega_d = 3 \times 2^{d/2-1}.$$

## 3 Results

We now give the main result of the paper: an oracle inequality for the estimator $\hat{\Delta}_{\hat{h}}$, or in other words, a bound on the risk that shows that the performance of the estimator is almost as good as it would be if we knew the risks of each estimator. In particular it performs an (almost) optimal trade-off between the variance term $V(h)$ and the approximation term

$$D(h) = D_f(h) = \max\left\{ \|(p\Delta_{\mathrm{P}} - \mathbb{E}[\hat{\Delta}_h])f\|_{2,M}, \sup_{h'\leq h}\|(\mathbb{E}[\hat{\Delta}_{h'}] - \mathbb{E}[\hat{\Delta}_h])f\|_{2,M}\right\}$$

$$\leq 2\sup_{h'\leq h}\|(p\Delta_{\mathrm{P}} - \mathbb{E}[\hat{\Delta}_{h'}])f\|_{2,M}.$$

**Theorem 3.1.** *According to the notation introduced in the previous section, let $\epsilon = \sqrt{a}/2 - 1$ and*

$$\delta(h) = \sum_{h'\leq h}\max\left\{\exp\left(-\frac{\min\{\epsilon^2,\epsilon\}\sqrt{n}}{24}\right), \exp\left(-c\epsilon^2\gamma_d(h')\right)\right\}$$

*where $c > 0$ is an absolute constant and*

$$\gamma_d(h') = \frac{1}{\|p\|_\infty h'^d}\left[\frac{2\omega_d \|K_d\|_{2,d}^2 + \alpha_d(h')}{(2\omega_d \|K_d\|_{1,d} + \beta_d(h))^2}\right]$$

*with $\alpha_d$ defined by (8) and*

$$\beta_d(h) = 2h\omega_d \|K_d\|_{1,d} + h^2(2\tilde{D}_3 + D) + h^3(\tilde{D}_4 + D). \tag{9}$$

*Given $f \in \mathcal{C}^2(M, \mathbb{R})$, with probability at least $1 - 2\sum_{h\in\mathcal{H}}\delta(h)$,*

$$\|(p\Delta_{\mathrm{P}} - \hat{\Delta}_{\hat{h}})f\|_{2,M} \leq \inf_{h\in\mathcal{H}}\left\{3D(h) + (1+\sqrt{2})\sqrt{bV(h)}\right\}.$$

Broadly speaking, Theorem 3.1 says that there exists an event of large probability for which the estimator selected by Lepski's method is almost as good as the best estimator in the collection. Note that the size of the bandwith family $\mathcal{H}$ has an impact on the probability term $1 - 2\sum_{h\in\mathcal{H}}\delta(h)$. If $\mathcal{H}$ is not too large, an oracle inequality for the risk of $\hat{\Delta}_{\hat{h}}f$ can be easily deduced from the later result. Henceforth we assume that $f \in \mathcal{F}$. We first give a control on the approximation term $D(h)$.

**Proposition 3.2.** *Assume that the density $p$ is $\mathcal{C}^2$. It holds that*

$$D(h) \leq \gamma\, C_{\mathcal{F}}h$$

*where $C_{\mathcal{F}}$ is defined in eq. (5) and $\gamma > 0$ is a constant depending on $\|p\|_\infty$, $\|p'\|_\infty$, $\|p''\|_\infty$ and on $M$.*

We consider the following grid of bandwidths:

$$\mathcal{H} = \left\{e^{-k}\,,\, \lceil\log\log(n)\rceil \leq k \leq \lfloor\log(n)\rfloor\right\}.$$

The previous results lead to the pointwise rate of convergence of the graph Laplacian selected by Lepski's method:

**Theorem 3.3.** *Assume that the density $p$ is $\mathcal{C}^2$. For any $f \in \mathcal{F}$, we have*

$$\mathbb{E}\left[\|(p\Delta_{\mathrm{P}} - \hat{\Delta}_{\hat{h}})f\|_{2,M}\right] \lesssim n^{-\frac{1}{d+4}}. \tag{10}$$

# 4 Sketch of the proof of theorem 3.1

We observe that the following inequality holds

$$\|(p\Delta_{\mathrm{P}} - \hat{\Delta}_{\hat{h}})f\|_{2,M} \le D(h) + \|(\mathbb{E}[\hat{\Delta}_h] - \hat{\Delta}_h)f\|_{2,M} + \sqrt{2\left(B(h) + bV(h)\right)}. \qquad (11)$$

Indeed, for $h \in \mathcal{H}$,

$$\|(p\Delta_{\mathrm{P}} - \hat{\Delta}_{\hat{h}})f\|_{2,M} \le \|(p\Delta_{\mathrm{P}} - \Delta_h)f\|_{2,M} + \|(\Delta_h - \hat{\Delta}_h)f\|_{2,M} + \|(\hat{\Delta}_h - \hat{\Delta}_{\hat{h}})f\|_{2,M}$$
$$\le D(h) + \|(\Delta_h - \hat{\Delta}_h)f\|_{2,M} + \|(\hat{\Delta}_h - \hat{\Delta}_{\hat{h}})f\|_{2,M}.$$

By definition of $B(h)$, for any $h' \le h$,

$$\|(\hat{\Delta}_{h'} - \hat{\Delta}_h)f\|_{2,M}^2 \le B(h) + aV(h') \le B(\max\{h, h'\}) + aV(\min\{h, h'\}),$$

so that, according to the definition of $\hat{h}$ in eq. (3) and recalling that $a \le b$,

$$\|(\hat{\Delta}_{\hat{h}} - \hat{\Delta}_h)f\|_{2,M}^2 \le 2\left[B(h) + aV(h)\right] \le 2\left[B(h) + bV(h)\right]$$

which proves eq. (11).

We are now going to bound the terms that appear in eq. (11). The bound for $D(h)$ is already given in proposition 3.2, so that in the following we focus on $B(h)$ and $\|(\mathbb{E}[\hat{\Delta}_h] - \hat{\Delta}_h)f\|_{2,M}$. More precisely the bounds we present in the next two propositions are based on the following lemma from [LM15].

**Lemma 4.1.** *Let* $X_1, \dots, X_n$ *be an i.i.d. sequence of variables. Let* $\widetilde{\mathcal{S}}$ *a countable set of functions and let* $\eta(s) = \frac{1}{n}\sum_i \left[g_s(X_i) - \mathbb{E}[g_s(X_i)]\right]$ *for any* $s \in \widetilde{\mathcal{S}}$. *Assume that there exist constants* $\theta$ *and* $v_g$ *such that for any* $s \in \widetilde{\mathcal{S}}$

$$\|g_s\|_\infty \le \theta \quad and \quad \mathrm{Var}[g_s(X)] \le v_g.$$

*Denote* $H = \mathbb{E}[\sup_{s \in \widetilde{\mathcal{S}}} \eta(s)]$. *Then for any* $\epsilon > 0$ *and any* $H' \ge H$

$$\mathbb{P}\left[\sup_{s \in \widetilde{\mathcal{S}}} \eta(s) \ge (1+\epsilon)H'\right] \le \max\left\{\exp\left(-\frac{\epsilon^2 n H'^2}{6 v_g}\right), \exp\left(-\frac{\min\{\epsilon, 1\}\epsilon n H'}{24\,\theta}\right)\right\}.$$

**Proposition 4.2.** *Let* $\epsilon = \frac{\sqrt{a}}{2} - 1$. *Given* $h \in \mathcal{H}$, *define*

$$\delta_1(h) = \sum_{h' \le h} \max\left\{\exp\left(-\frac{\min\{\epsilon^2, \epsilon\}\sqrt{n}}{24}\right), \exp\left(-\frac{2\epsilon^2}{3}\gamma_d(h')\right)\right\}.$$

*With probability at least* $1 - \delta_1(h)$

$$B(h) \le 2D(h)^2.$$

**Proposition 4.3.** *Let* $\tilde{\epsilon} = \sqrt{a} - 1$. *Given* $h \in \mathcal{H}$ *define*

$$\delta_2(h) = \max\left\{\exp\left(-\frac{\min\{\tilde{\epsilon}^2, \tilde{\epsilon}\}\sqrt{n}}{24}\right), \exp\left(-\frac{\tilde{\epsilon}^2}{24}\gamma_d(h)\right)\right\}.$$

*With probability at least* $1 - \delta_2(h)$

$$\|(\mathbb{E}[\hat{\Delta}_h] - \hat{\Delta}_h)f\|_{2,M} \le \sqrt{aV(h)}.$$

Combining the above propositions with eq. (11), we get that, for any $h \in \mathcal{H}$, with probability at least $1 - (\delta_1(h) + \delta_2(h))$,

$$\|(p\Delta_{\mathrm{P}} - \hat{\Delta}_{\hat{h}})f\|_{2,M} \le D(h) + \sqrt{aV(h)} + \sqrt{4D(h)^2 + 2bV(h)}$$
$$\le 3D(h) + (1+\sqrt{2})\sqrt{bV(h)}$$

where we have used the fact that $a \le b$. Taking a union bound on $h \in \mathcal{H}$ we conclude the proof.

# 5 Numerical illustration

In this section we illustrate the results of the previous section on a simple example. In section 5.1, we describe a practical procedure when the data set $\mathbb{X}$ is sampled according to the uniform measure on $M$. A numerical illustration us given in Section 5.2 when $M$ is the unit 2-dimensional sphere in $\mathbb{R}^3$.

## 5.1 Practical application of the Lepksi's method

Lepski's method presented in Section 2 can not be directly applied in practice for two reasons. First, we can not compute the $L^2$-norm $\| \ \|_{2,M}$ on $M$, the manifold $M$ being unknown. Second, the variance terms involved in Lepski's method are not completely explicit.

Regarding the first issue, we can approximate $\| \ \|_{2,M}$ by splitting the data into two samples: an estimation sample $\mathbb{X}_1$ for computing the estimators and a validation sample $\mathbb{X}_2$ for evaluating this norm. More precisely, given two estimators $\hat{\Delta}_h f$ and $\hat{\Delta}_{h'} f$ computed using $\mathbb{X}_1$, the quantity $\|(\hat{\Delta}_h - \hat{\Delta}_{h'})f\|_{2,M}^2/\mu(M)$ is approximated by the averaged sum $\frac{1}{n_2} \sum_{x \in \mathbb{X}_2} |\hat{\Delta}_h f(x) - \hat{\Delta}_{h'} f(x)|^2$, where $n_2$ is the number of points in $\mathbb{X}_2$. We use these approximations to evaluate the bias terms $B(h)$ defined by (4).

The second issue comes from the fact that the variance terms involved in Lepski's method depend on the metric properties of the manifold and on the sampling density, which are both unknown. Theses variance terms are thus only known up to a multiplicative constant. This situation contrasts with more standard frameworks for which a tight and explicit control on the variance terms can be proposed, as in [Lep92b, Lep93, Lep92a]. To address this second issue, we follow the calibration strategy recently proposed in [LM15] (see also [LMR16]). In practice we remove all the multiplicative constants from $V(h)$: all these constants are passed into the terms a and b. This means that we rewrite Lepski's method as follows:

$$\hat{h}(a,b) = \arg\min_{h \in \mathcal{H}} \left\{ B(h) + b\frac{1}{nh^4} \right\}$$

where

$$B(h) = \max_{h' \leq h, \, h' \in \mathcal{H}} \left[ \|(\hat{\Delta}_{h'} - \hat{\Delta}_h)f\|_{2,M}^2 - a\frac{1}{nh'^4} \right]_+ .$$

We choose $a$ and $b$ according to the following heuristic:

1. Take $b = a$ and consider the sequence of selected models: $\hat{h}(a,a)$,
2. Starting from large values of $a$, make $a$ decrease and find the location $a_0$ of the main *bandwidth jump* in the step function $a \mapsto \hat{h}(a,a)$,
3. Select the model $\hat{h}(a_0, 2a_0)$.

The justification of this calibration method is currently the subject of mathematical studies ([LM15]). Note that a similar strategy called "slope heuristic" has been proposed for calibrating $\ell_0$ penalties in various settings by strong mathematical results, see for instance [BM07, AM09, BMM12].

## 5.2 Illustration on the sphere

In this section we illustrate the complete method on a simple example with data points generated uniformly on the sphere $\mathbb{S}^2$ in $\mathbb{R}^3$. In this case, the weighted Laplace-Beltrami operator is equal to the (non weighted) Laplace-Beltrami operator on the sphere.

We consider the function $f(x,y,z) = (x^2 + y^2 + z)\sin x \cos x$. The restriction of this function on the sphere has the following representation in spherical coordinates:

$$\tilde{f}(\theta, \phi) = (\sin^2 \phi + \cos \phi)\sin(\sin \phi \cos \theta)\cos(\sin \phi \cos \theta).$$

It is well known that the Laplace-Beltrami operator on the sphere satisfies (see Section 3 in [Gri09]):

$$\Delta_{\mathcal{S}^2} u = \frac{1}{\sin^2 \phi} \frac{\partial^2 u}{\partial \theta^2} + \frac{1}{\sin \phi} \frac{\partial}{\partial \phi} \left( \sin \phi \frac{\partial u}{\partial \phi} \right)$$

for any smooth polar function $u$. This allows us to derive an analytic expression of $\Delta_{\mathcal{S}^2} \tilde{f}$.

We sample $n_1 = 10^6$ points on the sphere for computing the graph Laplacians and we use $n = 10^3$ points for approximating the norms $\|(\hat{\Delta}_h - \hat{\Delta}_{h'})\tilde{f}\|_{2,M}^2$. We compute the graph Laplacians for bandwidths in a grid $\mathcal{H}$ between 0.001 and 0.8 (see fig. 1). The risk of each graph Laplacian is estimated by a standard Monte Carlo procedure (see fig. 2).

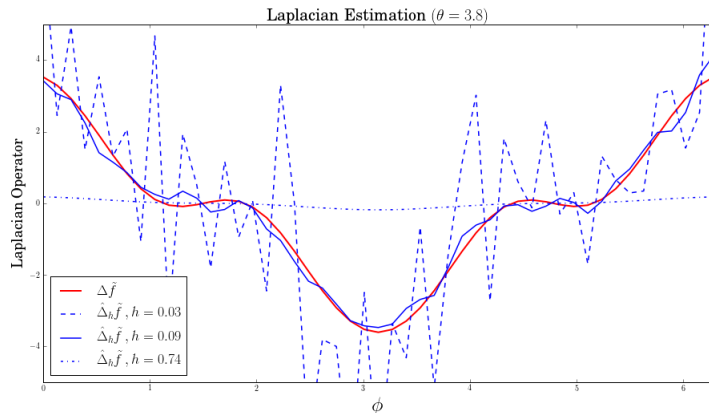

Figure 1: Choosing $h$ is crucial for estimating $\Delta_{\mathcal{S}^2}\tilde{f}$: small bandwidth overfits $\Delta_{\mathcal{S}^2}\tilde{f}$ whereas large bandwidth leads to almost constant approximation functions of $\Delta_{\mathcal{S}^2}\tilde{f}$.

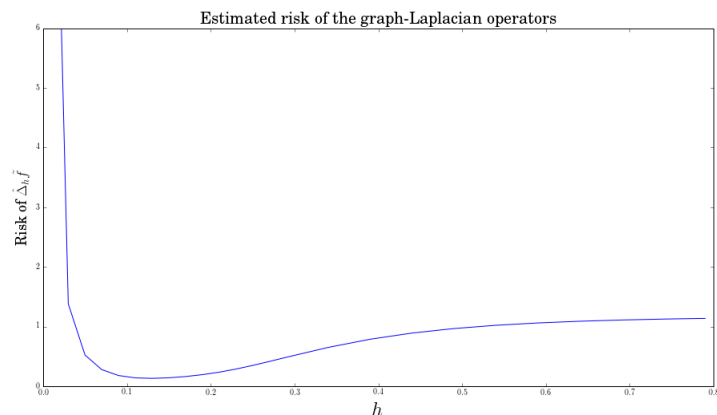

Figure 2: Estimation of the risk of each graph Laplacian operator: the oracle Laplacian is for approximatively $h = 0.15$.

Figure 3 illustrates the calibration method. On this picture, the $x$-axis corresponds to the values of $a$ and the $y$-axis represents the bandwidths. The blue step function represents the function $a \mapsto \hat{h}(a,a)$. The red step function gives the model selected by the rule $a \mapsto \hat{h}(a,2a)$. Following the heuristics given in Section 5.1, one could take for this example the value $a_0 \approx 3.5$ (location of the bandwidth jump for the blue curve) which leads to select the model $\hat{h}(a_0, 2a_0) \approx 0.2$ (red curve).

## 6  Discussion

This paper is a first attempt for a complete and well-founded data driven method for inferring Laplace-Beltrami operators from data points. Our results suggest various extensions and raised some questions of interest. For instance, other versions of the graph Laplacian have been studied in the literature (see

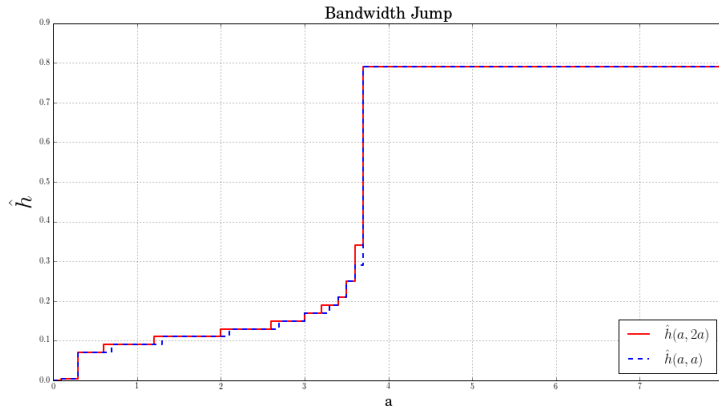

Figure 3: Bandwidth jump heuristic: find the location of the jump (blue curve) and deduce the selected bandwidth with the red curve.

for instance [HAL07, BN08]), for instance when data is not sampled uniformly. It would be relevant to propose a bandwidth selection method for these alternative estimators also.

From a practical point of view, as explained in section 5, there is a gap between the theory we obtain in the paper and what can be done in practice. To fill this gap, a first objective is to prove an oracle inequality in the spirit of Theorem 3.1 for a bias term defined in terms of the empirical norms computed in practice. A second objective is to propose mathematically well-founded heuristics for the calibration of the parameters $a$ and $b$.

Tuning bandwidths for the estimation of the spectrum of the Laplace-Beltrami operator is a difficult but important problem in data analysis. We are currently working on the adaptation of our results to the case of operator norms and spectrum estimation.

## Appendix: the geometric constants $C$ and $C_1$

The following classical lemma (see, e.g. [GK06][Prop. 2.2 and Eq. 3.20]) relates the constants $C$ and $C_1$ introduced in Equations (6) and (7) to the geometric structure of $M$.

**Lemma 6.1.** *There exist constants $C, C_1 > 0$ and a positive real number $r > 0$ such that for any $x \in M$, and any $v \in T_x M$ such that $\|v\| \leq r$,*

$$\left| \sqrt{\det(g_{ij})}(v) - 1 \right| \leq C_1 \|v\|_d^2 \qquad and \qquad \frac{1}{2} \|v\|_d^2 \leq \|v\|_d^2 - C\|v\|_d^4 \leq \|\mathcal{E}_x(v) - x\|_m^2 \leq \|v\|_d^2$$

(12)

*where $\mathcal{E}_x : T_x M \to M$ is the exponential map and $(g_{i,j})_{i,j} \in \{1, \cdots, d\}$ are the components of the metric tensor in any normal coordinate system around $x$.*

Although the proof of the lemma is beyond the scope of this paper, notice that one can indeed give explicit bounds on $r$ and $C$ in terms of the reach and injectivity radius of the submanifold $M$.

### Acknowledgments

The authors are grateful to Pascal Massart for helpful discussions on Lepski's method. This work was supported by the ANR project TopData ANR-13-BS01-0008 and ERC Gudhi No. 339025

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
