[Reviews · NeurIPS 2016]

Reviewer 1

Summary

This paper provides a method for choosing the bandwidth parameter of a point cloud Laplacian (but introduces other parameters in its place). Theoretical guarantees and a minimalistic numerical experiment are provided as evidence of usefulness of the technique.

Qualitative Assessment

I am unqualified to assess the theoretical significance of this paper but am quite concerned about the practicality. My key concern about this paper is that it does not seem to improve the state of affairs re: parameters needed for Laplacian approximations. In particular, the abstract/intro led me to believe the paper was going to alleviate the need to choose a bandwidth parameter for the Laplacian (a very worthy task!). But in reality, the technique still has parameters a, b, and the paper says "calibration of the constants a and b in practice is beyond the scope" (line 92). If this is the case, what is the value of the current paper? The restrictions described at the beginning of sec 5.1 are similarly concerning. Beyond this, the numerical experiments are far too minimal. Recovering the Laplacian of a sphere does not represent the noise and geometry typically encountered in machine learning. Other comments: - There are quite a few spelling mistakes and other typos that should be fixed; rather than list them here I will encourage the authors to run "spell check" - You can say "Lepski's method" instead of "the Lepski's method" - Say early on (e.g. in sec 1) whether the dimension d should be known a priori for this method to be useful. (Perhaps a relevant citation is "Testing the Manifold Hypothesis" by Fefferman and colleagues) - l. 32 -- cite papers instead of saying "known results" - l. 39 -- define "contrast" - The assumption that H is finite (line 67) seems quite limiting and does not align with e.g. choosing the width of a Gaussian kernel - I am a little confused about the density function P discussed around eq (2). In particular, how does P interact with the parameterization chosen in eq (1)? Is it possible for the isotropic kernel in eq (1) to deal with irregular or unevenly-distributed P's? - The explanation at the beginning of sec 2.1 of Lepski's method was *very* dense for an inexperienced reader. Please add equations and/or a concrete example rather than just describing from a high level in words. - l. 79 "thus seems natural" -- why? - In eq (3), it seems that h depends on f. Is this right? If so, does that make the Laplacian nonlinear? - In prop 2.1, I am unclear what it means to have a formula for V(h). It appears above that V(h) is *any* bound satisfying the condition in line 89. Is this an upper bound for V(h)? A formula that satisfies the definition? - The heuristic in lines 170-177 comes out of thin air. Please provide justification or at least intuition.

Confidence in this Review

1-Less confident (might not have understood significant parts)


Reviewer 2

Summary

The derivation is essentially an extension/application of [LM15] to the bandwidth estimation problem in the construction of the Laplace-Beltrami operator from random samples on a smooth manifold. In particular, the paper extends the Lepski's method, similarly to [LM15], and apply it to the specific problem of the Laplacian operator estimation. In practice, very much like [LM15], it compute heuristically two parameters a and b, that serve to avoid the explicit computation of the variance terms in the Lepski method. The results illustrate the influence the proper choice of the bandwidth in the LB estimation on a spherical manifold.

Qualitative Assessment

The paper is mostly theoretical, and addresses an important problem. However, due to the different approximation and heuristics, it is hard to understand if the proposed methods and the resulting bounds are indeed meaningful. A more extensive discussions of the benefits and importance of the proposed results, and possibly the application of the method to different objects and not only the sphere would be interesting. Finally, a better positioning with respect to the numerous works that study the estimation of the LB operators would be good, even numerically when appropriate. What does one really gain by using the proposed bw estimation method? And when? For what type of manifold/problem? Responses to such questions would help making the paper more relevant, and certainly its nice results more visible. Finally, non-uniform sampling is not discussed unfortunately, despite what is announced in the text. The Appendix could probably be put in the text, that would certainly not penalise the clarity of the development.

Confidence in this Review

2-Confident (read it all; understood it all reasonably well)


Reviewer 3

Summary

The authors suggested estimator for Laplace-Beltrami operator. The Laplace-Beltrami operator carries topological and geometrical information of the manifold. The authors used nonparametric graph Laplacians to estimate Laplace-Beltrami operator. For choosing the bandwidth h, the authors used Lepski's procedure. The authors ran their method on the simulated sphere.

Qualitative Assessment

This paper shows nice theoretical risk analysis for the suggested Laplace-Beltrami estimator. The paper is nicely organized and is pleasant to be read. I have two comments/questions: 1. It would be nice if more motivations for Laplace-Beltrami operator are provided, such as, what kind of information can be carried on by Laplace-Beltrami operators, or how it is useful for analyzing data. 2. Is there any motivation or reference for the heuristic of choosing a and be in Line 171-174?

Confidence in this Review

2-Confident (read it all; understood it all reasonably well)


Reviewer 4

Summary

When data comes from a manifold, we can ask questions about the relationship between Laplacian operators on the graph of the data and the Lapace-Beltrami operator on the underlying manifold. A key challenge in studying a graph Laplacian is parameter tuning, and in particular turning the bandwidth parameter is studied here. The authors give a review of Lepski's method for estimation of the Laplace-Beltrami operator, which chooses the estimator from a finite collection of estimators which roughly minimizes a linear combination of bias and variance, where the bias is estimated by maximizing over the estimators with smaller bandwidth. The authors discuss why this method needs adaptation to the current problem (data from manifolds), and then compute variances of graph Laplacians for smooth functions, meeting one of the conditions for the use of Lepski's method. The crux of the remaining problem is that the risk of an estimator cannot be computed, preventing the use of Lepski's method. The authors develop an oracle inequality which leads to a bound that Lepski's method using the oracle inequality is likely to be near optimal. The authors discuss that in practice, variance terms and the norm of the Manifold underlying the data are not available explicitly, and so they discuss heuristics for approximating these, as well as a heurstic suggested by Lepski for computing the weights of the bias and variance terms, yielding a useable version of Lepski's method in their problem statement. They conclude with an illustration of their adaption of the method to the sphere using a function with a known spherical coordinate representation. Their method's estimation is comparted to the known Laplace-Beltrami operator on the sphere. They demonstrate how the bias variance tradeoff is handled well by the method.

Qualitative Assessment

The paper was well written in general and did an excellent job of balancing out the highly technical material with intuitive explanations. I would have like a little more contextualization of the applications of the studied problem and why estimations of the operator itself will improve data analysis beyond its discretizations. My major concern with the paper is that there are still holes in the adaptation of the algorithm. The work here contributes to a finished algorithm by a theory bound in one of the challenges of adapting the method to data, but there are still holes in computing a,b, and the 2,M norm. The result discusses heuristics and shows that the method works well in the synthetic realm, but I think in the absence of a complete theoretical picture, the work would be well served by some empirical applications.

Confidence in this Review

1-Less confident (might not have understood significant parts)


Reviewer 5

Summary

The authors propose a method for choosing the bandwidth when constructing approximations to the Laplace-Beltrami operator using points sampled from a manifold. It applies a modified version of Lepski's method which this reviewer understands as a method of estimating the bias term when the variance term can be treated as effectively known known and choosing the corresponding optimal bandwidth. The authors show that the unmodified Lepski's method applied to the bandwidth selection problem

Qualitative Assessment

This is a nice contribution for describing a way to choose the bandwidth for Laplacians constructed from point clouds. However, the gap in the theory where a heuristic is required to choose the multipliers a & b makes this reviewer somewhat uneasy since the multiplicative constant can change the variance estimate by an arbitrary amount. This makes the final algorithm seem like a well-motivated rule of thumb for choosing the bandwidth. The authors do, however, mention that the heuristic is being studied and a similar method has strong mathematical results. The large bandwidth jump appears interesting and allays some of the concerns, but without an explanation why a large jump should be expected, one example is not enough to lead a reader to believe that this is a normal phenomenon. The main question for the authors is why does the variance term need to have a multiplicative constant? Is there a reason a resampling method can't estimate the variance term? If it can, that would seem to remove concerns about the gaps in the theory, and the much simpler to compute heuristic approach can be empirically compared to it. It would also be nice to see 1 or 2 more empirical examples, especially on an unsupervised / semi-supervised application of a Laplacian based method where the bandwidth choice cannot be easily obtained with cross validation, e.g. Laplacian eigenmaps or spectral clustering. The previously proven convergence guarantees have only been for manifolds without boundary or functions with support on the interior of the manifold, but a practical bandwidth selection technique would need to handle boundaries. Please make clear in the Prop 2.1 itself that V(h) is a variance bound rather than the variance. Does eps need to be > 0 so alpha > 4 in the proof? It seems a little odd since the most natural choice of alpha appears to be 2 from the description of Lepski's method. ln 136. and -> be an What are theta and v in Lemma 4.1? Having both nu and v appear at the same place is less than ideal. ln 174 a -> a_0?

Confidence in this Review

3-Expert (read the paper in detail, know the area, quite certain of my opinion)